# AC-ODM: Actor–Critic Online Data Mixing for Sample-Efficient LLM Pretraining

## Abstract

Pretraining data coverage and composition strongly influence the generalization of large language models (LLMs). While recent data-mixing approaches transfer domain weights learned by a small proxy model to a larger one to reduce computational costs and carbon footprint, they are typically static and ignore training dynamics. Online Data Mixing (ODM) mitigates this with a multi-armed bandit sampler but overlooks intra-domain interactions. We introduce AC-ODM, an actor–critic online data-mixing method that treats the LLM as the environment, uses auxiliary actor–critic networks to dynamically adjust domain sampling weights, and encodes intra-domain interactions through the reward. AC-ODM supports (i) a non-proxy mode that co-trains the actor–critic with the target LLM from scratch, and (ii) a proxy mode that first trains the actor–critic with a small, trainable proxy LLM and then transfers the learned actor to guide the target LLM's pretraining. Empirically, the proxy mode incurs additional wall-clock time relative to the non-proxy mode but delivers stronger target-LLM performance. Across both modes, AC-ODM enables efficient, adaptive data mixing and accelerates target-model convergence, with negligible per-step wall-clock overhead. On Pythia-1B pretraining over The Pile and SlimPajama, AC-ODM-410M (a policy learned with a 410M-parameter proxy) reaches the optimal validation perplexity of ODM using 71% and 65% fewer training steps, respectively. It achieves a 27.5% relative improvement in zero-shot MMLU accuracy, a $2.23\times$ higher pass@1 on HumanEval, and an average +3.44% accuracy gain across five additional benchmarks. We further show that AC-ODM maintains the fastest pretraining convergence on LLaMA3-style architectures compared to prior data-mixing baselines.

## 1 Introduction

The pretraining corpus is a primary determinant of the generalization ability of large language models (LLMs). Its coverage and composition strongly influence both sample efficiency and downstream accuracy John & Draper (1975); Du et al. (2022). When the corpus is fixed, the quantity and mixture of selected data largely determine how much useful information the model can absorb as well as the convergence speed during pretraining Lee et al. (2022); Sorscher et al. (2023); Xie et al. (2023b); Albalak et al. (2024). Nevertheless, domain weights for many state-of-the-art LLMs are chosen by heuristics Gao et al. (2020), which raises the question of whether a more effective set of weights can be learned.

Data mixing methods aim to optimize domain weights to improve training efficiency and final performance Xie et al. (2023a); Fan et al. (2024); Albalak et al. (2023); Xia et al. (2024). Broadly, methods either determine the weights before model training in an offline manner or adapt the weights during training in an online manner Albalak et al. (2024). The representative offline approach DoReMi Xie et al. (2023a) adopts a two stage pipeline. A small reference model is first trained with uniform domain weights, then a proxy model is trained to maximize the information gain over the reference and the learned domain weights are transferred to the target model as sampling probabilities. In addition to training two auxiliary models, the objective of minimizing the worst case loss gap is not perfectly aligned with producing a well trained target model. Following this route, DoGE Fan et al. (2024) optimizes gradient alignment between each domain and the remainder of the corpus rather than excess loss. However, Xie et al. (2023a) showed that the weights obtained in this two stage pipeline transfer poorly across architectures and tokenizers because they do not adapt to the

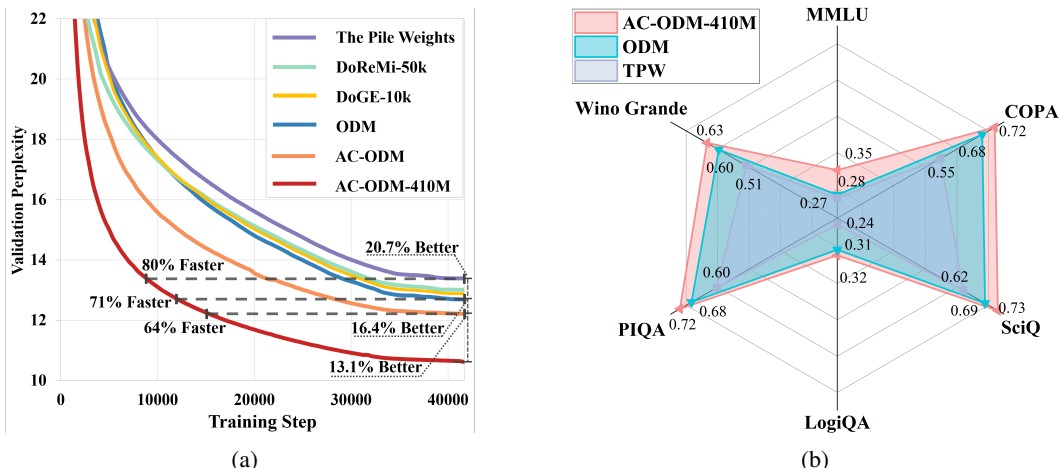

(a)

(b)

Figure 1: **Training dynamics and downstream generalization.** **(a)** Validation perplexity during Pythia-1B pretraining on The Pile, reported as the unweighted average across its 22 domains. We compare The Pile weights(TPW), DoReMi-50k, DoGE-10k, ODM, AC-ODM (non-proxy), and AC-ODM-410M (proxy). **(b)** 5-shot accuracy radar over downstream benchmarks. AC-ODM-410M consistently surpasses ODM and TPW.

changing dynamics of pretraining. As a result, a new model often needs to be trained to rediscover appropriate weights for each setting, which compromises both the effectiveness and the efficiency of the offline framework.

To track the training dynamics of the target model directly, Online Data Mixing (ODM) Albalak et al. (2023) formulates domain selection as a multi armed bandit problem based on EXP3 Auer et al. (2002). ODM treats each domain as an arm and uses the instantaneous loss as the reward, which biases sampling toward domains that currently incur larger losses. This encourages learning from domains whose data distributions are farther from the model's predictions. At the same time, this reward assumes uniform data quality and overlooks rich intra domain interactions. In practice, examples from one domain can accelerate learning on other domains because of lexical, syntactic, or semantic overlap. Emphasizing domains with strong cross domain interactions at early stages could therefore speed up pretraining.

We propose **AC-ODM**, an actor–critic online data mixing method that models the impact of domain weights on LLM pretraining as a reinforcement learning problem. Inspired by DoGE, we define the reward using a gradient alignment measure $W$, which captures both loss reduction and intra domain interactions. Maximizing the expected reward prioritizes datasets with strong commonalities and more general utility, which accelerates convergence. We adopt the deep deterministic policy gradient algorithm to learn continuous sampling weights. The LLM serves as the environment, and lightweight actor and critic networks are trained to produce domain weights conditioned on the current training state. AC-ODM supports two operational modes. In the *non proxy* mode, the actor–critic is co trained with the target LLM from scratch, which adds negligible per step wall clock overhead. In the *proxy* mode, the actor–critic is first trained with a small trainable proxy LLM and the learned actor is then transferred to guide dynamic sampling for the target LLM. The proxy mode introduces additional wall clock time relative to the non proxy mode but yields stronger target model performance in our experiments. As shown in Fig. 1a, AC-ODM markedly accelerates pretraining by reaching the optimal validation perplexity of ODM in fewer steps. Complementarily, Fig. 1b shows consistent gains on downstream evaluation, including higher 5-shot MMLU accuracy, which evidences the effectiveness of our reward and state designs.

The main contributions of this work are two fold.

1. We propose a actor-critic based online data mixing approach that explicitly models intra domain interactions. Our central novelty is a gradient alignment reward that measures how updates from one domain accelerate progress on others, so the policy prioritizes domains

that most benefit the whole corpus. Treating the LLM under training as the environment, we design a compact state that captures its evolving status and conditions the policy on the current dynamics. AC-ODM steers the pretraining process towards faster convergence and better generalization.

2. We improve the efficiency and practicality of online data mixing with two complementary modes. The non proxy mode enables end to end adaptive mixing with negligible per step overhead. The proxy mode learns a policy with a 410M parameter proxy and transfers the actor to a 1B target model, which accelerates convergence and strengthens downstream generalization. On Pythia 1B pretraining over The Pile and SlimPajama, AC-ODM-410M reaches the optimal validation perplexity of ODM using $71\%$ and $65\%$ fewer steps, respectively. It delivers a $27.5\%$ relative improvement in zero shot MMLU accuracy, a $2.23\times$ higher pass@1 on HumanEval, and an average $+3.44\%$ accuracy gain across five additional benchmarks, and it maintains the fastest pretraining convergence on LLaMA3 style architectures.

## 2 AC-ODM

In this section, we present actor critic online data mixing (AC-ODM) for efficient and adaptive pretraining of large language models.

### 2.1 PROBLEM FORMULATION

Let $D = \{D_1, \ldots, D_k\}$ be a corpus composed of $k$ domains for language model pretraining. We seek domain weights on the probability simplex $\alpha \in \Delta^k \subset \mathbb{R}^k$. Training batches are produced by first sampling a domain according to the domain wise distribution $\alpha$ and then sampling uniformly within that domain, namely $B \sim \mathrm{UNIF}(D_i)$. This induces the instance wise distribution $P_\alpha \triangleq \sum_{i=1}^k \alpha_i \cdot \mathrm{UNIF}(D_i)$. Offline data mixing fixes $P_\alpha$ before training, while online data mixing updates $P_\alpha$ at every iteration. Our objective is to adapt $P_\alpha$ online with negligible wall clock overhead.

### 2.2 ADAPTING ACTOR CRITIC TO ONLINE DATA MIXING

We cast online data mixing as a Markov decision process and adopt the deep deterministic policy gradient framework. As illustrated in Figure 2, the LLM defines the environment with state $s^t$. The agent executes action $a^t = \mu(s^t)$ that updates the domain weights $\alpha$ and therefore specifies $P_\alpha$. At each step $t \in \{1, \ldots, T\}$, the state aggregates observable training signals of the LLM, including the number of samples per domain $n = \{n_i\}_{i=1}^k$, the iteration index $t$, the per domain loss $\ell(\theta_M, B) = \{\ell_i(\theta_M, B_i)\}_{i=1}^k$ and its difference with the previous step $\Delta\ell(\theta_M, B) = \{\Delta\ell_i(\theta_M, B_i)\}_{i=1}^k$, as well as the $L_2$ norm of selected LLM layer weights $\|\omega\|_2$ and the step to step change $\|\Delta\omega\|_2$. The agent maximizes the expected return $\mathbb{E}[\sum_{t=0}^T \gamma^t r^t]$ by adjusting $\alpha_i^t \rightarrow \alpha_i^{t+1}$, training the LLM on the sampled batch, and observing the next state $s^{t+1}$. Since both the state and the action are continuous, DDPG is a suitable optimizer.

Formally, the agent environment tuple is defined as follows. The state is $s^t = (n, t, \ell(\theta_M, B), \Delta\ell(\theta_M, B), \|\omega\|_2, \|\Delta\omega\|_2)$. The action is $a^t = [a_1^t, \ldots, a_k^t]$. The reward vector is $r^t = [r_1^t, \ldots, r_k^t]$. The deterministic policy is $\mu_{\theta_A} : s^t \mapsto a^{t+1}$.

### 2.3 DESIGNING THE REWARD FUNCTION

Efficient and stable convergence requires a reward that values examples which accelerate learning in other domains while avoiding over concentration on a few domains. Inspired by DoGE Fan et al. (2024), we set the reward for domain $i$ to its gradient alignment score $W_i \triangleq \langle \nabla\ell_i(\theta_M), \sum_{j \in [k]} \nabla\ell_j(\theta_M) \rangle$, where $\nabla\ell_i(\theta_M)$ is the stochastic gradient for $B_i$. This score measures how much learning from domain $i$ supports progress on the rest of the corpus. We denote $W = [W_1, \ldots, W_k]$ and assign $r_i^t = W_i^t$. To smooth the signal, we maintain an exponential moving average with importance correction $\hat{r}_i^t = \xi \hat{r}_i^{t-1} + (1-\xi)\frac{r_i^t}{P_{\alpha_i}^{t-1}}$, where division by $P_{\alpha_i}^{t-1}$ discourages over sampling already frequent domains.

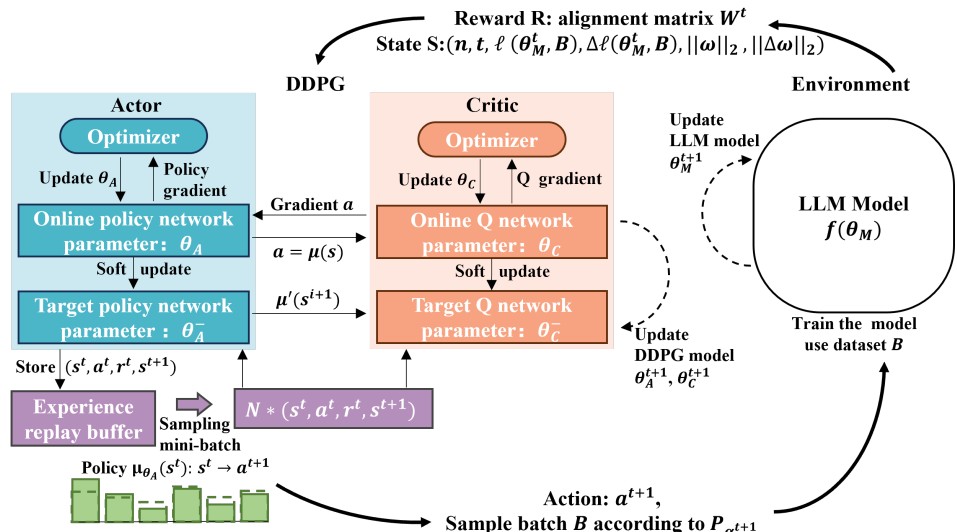

Figure 2: Overview of AC-ODM. At iteration $t$, the policy $\mu_{\theta_A}$ takes the environment state $s^t$ from the current LLM and outputs an action $a^t$ that adjusts the domain weights to $\alpha_t$. A batch $B$ is then sampled according to the instance wise distribution $P_\alpha$. The loss gradient $\nabla L(\theta_M^t, B)$ and the gradient alignment matrix $W^t$ are computed with respect to the model $f(\theta_M)$ to update $\theta_M$ and to produce the reward $r^t$. The transition to $s^{t+1}$ is recorded and the tuple $(s^t, a^t, r^t, s^{t+1})$ is stored in the replay buffer. The actor and critic parameters $\theta_A$ and $\theta_C$ are updated from $N$ samples drawn from the buffer. After $T$ steps, the learned online policy can be used directly in the non proxy mode or transferred from a proxy LLM to a target LLM in the proxy mode.

Computing $W$ can be expensive for large models and many domains due to memory traffic for gradients. AC-ODM mitigates this by allowing the policy to be learned once with a proxy LLM and then executed without reward computation during target model pretraining. In this way, the heavy part of reward estimation is confined to the proxy stage.

## 2.4 MODEL UPDATE

Each iteration updates three parameter sets: the LLM $\theta_M$, the critic $\theta_C$, and the actor $\theta_A$.

*Updating $\theta_M$.* Given $a^t$ and the induced weights $\alpha^t$, we sample $B$ according to $P_\alpha$ and compute the per domain losses and gradients. The proxy model is then updated with a loss reweighting factor $\alpha$:

$$\theta_M^{t+1} \triangleq \theta_M^t - \eta^t \sum_{i \in [k]} \alpha_i^t \nabla \ell_i(\theta_M^t).$$

*Updating $\theta_C$ and $\theta_A$.* Let the critic be $Q_{\theta_C}(s, a)$ and the actor be $\mu_{\theta_A}(s)$. We compute $r^t = W^t$ and the next state $s^{t+1} = (n^{t+1}, t+1, \ell(\theta_M^{t+1}, B), \Delta\ell(\theta_M^{t+1}, B), \|\omega^{t+1}\|_2, \|\Delta\omega^{t+1}\|_2)$, then store $(s^t, a^t, r^t, s^{t+1})$ in the replay buffer. For mini batch samples $\{(s^j, a^j, r^j, s^{j+1})\}_{j=1}^N$, the temporal difference target is

$$y_j = r^j + \gamma Q_{\theta_C}(s^{j+1}, \mu_{\theta_A}(s^{j+1})).$$

The critic minimizes

$$L = \frac{1}{N} \sum_{j=1}^N \left(y_j - Q_{\theta_C}(s^j, a^j)\right)^2.$$

The actor ascends the policy gradient

$$\nabla_{\theta_A} J \approx \frac{1}{N} \sum_{j=1}^N \nabla_{\theta_A} \mu_{\theta_A}(s^j) \nabla_a Q_{\theta_C}(s^j, a)\big|_{a = \mu_{\theta_A}(s^j)}.$$

We follow DDPG and maintain target networks for stability.

## 2.5 MODES OF AC-ODM

AC-ODM supports two operational modes that trade a small increase in wall clock time for stronger generalization.

---

**Algorithm 1** AC-ODM in the non proxy mode

---

**REQUIRE:** $D = \{D_1, \ldots, D_K\}$ grouped data
**REQUIRE:** $\theta_M^0$ target LLM weights, $\theta_A$ actor weights, $\theta_C$ critic weights
**REQUIRE:** $\nabla L_i(\theta_M^t)$ stochastic gradient of $B_i$ at step $t$
**REQUIRE:** Hyperparameters: total steps $T$, step size $\eta^t$, target update coefficient $\tau$, discount factor $\gamma$

1: Initialize $K = |D|$, set $r_i^0 = 0$ for all $i \in \{1, \ldots, K\}$, initialize critic $Q_{\theta_C}$, actor $\mu_{\theta_A}$, and LLM weights $\theta_M^0$
2: Copy target networks $\bar{\theta}_C \leftarrow \theta_C$, $\bar{\theta}_A \leftarrow \theta_A$
3: Initialize replay buffer $\mathcal{B}$, perform warm up to obtain the initial state $s^0 = (n^0, 0, \ell(\theta_M^0, B), \Delta\ell(\theta_M^0, B), \|\omega^0\|_2, \|\Delta\omega^0\|_2)$
4: **for** $t = 0$ to $T - 1$ **do**
5:     Choose action $a^t = \mu_{\theta_A}(s^t)$ and map to domain weights $\alpha^t$
6:     Sample batch $B^t = \{B_1^t, \ldots, B_K^t\}$ according to $P_\alpha \triangleq \sum_{i=1}^K \alpha_i^t \cdot \mathrm{UNIF}(D_i)$
7:     Compute $\nabla L_i(\theta_M^t)$ for all $i \in [K]$ and the alignment vector $W^t$
8:     Update the LLM: $\theta_M^{t+1} \leftarrow \theta_M^t - \eta^t \sum_{i=1}^K \alpha_i^t \nabla L_i(\theta_M^t)$
9:     Set $r^t \leftarrow W^t$
10:    Form the next state $s^{t+1} = (n^{t+1}, t + 1, \ell(\theta_M^{t+1}, B), \Delta\ell(\theta_M^{t+1}, B), \|\omega^{t+1}\|_2, \|\Delta\omega^{t+1}\|_2)$
11:    Store $(s^t, a^t, r^t, s^{t+1})$ in $\mathcal{B}$
12:    Sample $\{(s^j, a^j, r^j, s^{j+1})\}_{j=1}^N$ from $\mathcal{B}$
13:    Compute $y_j = r^j + \gamma Q_{\bar{\theta}_C}(s^{j+1}, \mu_{\bar{\theta}_A}(s^{j+1}))$
14:    Update critic by minimizing $L = \frac{1}{N} \sum_{j=1}^N (y_j - Q_{\theta_C}(s^j, a^j))^2$
15:    Update actor via $\nabla_{\theta_A} J \approx \frac{1}{N} \sum_{j=1}^N \nabla_{\theta_A} \mu_{\theta_A}(s^j) \nabla_a Q_{\theta_C}(s^j, a)\big|_{a = \mu_{\theta_A}(s^j)}$
16:    Soft update targets: $\bar{\theta}_A \leftarrow \tau\theta_A + (1 - \tau)\bar{\theta}_A$, $\bar{\theta}_C \leftarrow \tau\theta_C + (1 - \tau)\bar{\theta}_C$
17: **end for**
18: **return** actor $\mu_{\bar{\theta}_A}$

---

**Non proxy mode.** The actor and critic are trained jointly with the target LLM from scratch. This mode adds negligible per step overhead and yields fast convergence.

**Proxy mode.** The actor and critic are first trained with a smaller trainable proxy LLM using the same domains. The learned actor is then transferred to guide sampling for the target LLM. Reward computation and critic updates are performed only in the proxy stage, which reduces overhead during target pretraining and often produces stronger downstream performance.

---

**Algorithm 2** AC-ODM in the proxy mode

---

**REQUIRE:** Proxy LLM initialization $\theta_{M,\text{proxy}}^0$, target LLM initialization $\theta_{M,\text{tgt}}^0$, actor $\theta_A$, critic $\theta_C$, domains $D$

1: **Proxy stage:** Train the actor and critic with the proxy LLM using Algorithm 1 on $D$, obtain the trained actor $\mu_{\bar{\theta}_A}$
2: **Transfer:** Freeze the actor and remove reward computation
3: **Target stage:** For steps $t = 0$ to $T_{\text{tgt}} - 1$, sample batches for the target LLM with $\alpha^t = \mu_{\bar{\theta}_A}(s^t)$, update $\theta_{M,\text{tgt}}$ with the reweighted loss, and refresh the state $s^{t+1}$ as in Algorithm 1 without updating the actor or critic
4: **return** target LLM trained under the transferred actor policy

---

## 3 EXPERIMENTAL SETUP

We describe datasets, model training protocols, baseline configurations, and evaluation criteria. For the actor and critic networks, we also detail the state design and reward design. All experiments are

run on a single machine with an Intel(R) Xeon(R) Platinum 8468 CPU and 8 NVIDIA H800 GPUs with 80 GB memory each.

**LLM training**   We use The Pile Gao et al. (2020), an open source corpus of 825 GB from 22 diverse sources such as YouTube Subtitles, GitHub, and Wikipedia. In addition, we pretrain on SlimPajama Soboleva et al. (2023), a seven domain corpus containing 672B tokens at a smaller scale. Models are decoder only Transformers implemented with a modified GPT NeoX library Black et al. (2022). Unless noted otherwise, configurations follow Pythia Biderman et al. (2023) and we train a 1 billion parameter model. Each GPU processes a micro batch of 8 sequences. We use gradient accumulation across 8 GPUs with accumulation step 18, which yields an effective batch size of 1152 samples. For each batch, we first draw 10 percent per domain to expose the policy to intra domain relationships while preserving exploration and exploitation. The sequence length is 1024 with sequence packing Roberts et al. (2022). Training runs for 41,667 steps, corresponding to 50 billion tokens. During the first 833 warmup steps we replace AC driven weights with The Pile domain weights perturbed by Gaussian noise sampled from $N(0, 0.02)$.

**AC training**   The actor and critic share the same warmup and main training schedules as the LLM, with cosine decay learning rate starting at 0.01 and decaying to 0.001. During warmup, we train the actor and critic from the replay buffer $\mathcal{B}$. To initialize the actor, we use the LLM warmup domain weights as soft labels, namely the noisy The Pile weights, and optimize with mean squared error. For the critic, we initialize labels as $(1 + \gamma)r^t$ and optimize with mean squared error. During main training, each iteration samples 256 tuples from $\mathcal{B}$, dispatches 32 tuples per GPU, and uses gradient accumulation of 1, which gives an effective batch size of 256. Architectural details for Pythia 1B and the AC networks are in Appendix A.

**Reward and State setting**   For the state features in Pythia 1B, the term $\|\omega\|_2$ is computed on a subset of layers. We use the first Transformer layer together with all layers whose indices are even. This selection reduces computation time with negligible loss of fidelity. To balance efficiency and efficacy in reward computation, we restrict the calculation to a subset of parameters. Specifically, for Pythia 1B we use the final feedforward blocks of Transformer layers 12, 14, and 16, which together contain 50,331,648 parameters. This choice reduces memory traffic while preserving a faithful proxy for reward estimation. Ablations in Appendix B show that when only three layers are used this selection is optimal.

**Baselines**   We compare against the original The Pile domain weights Gao et al. (2020), referred to as The Pile Weights, and against domain weights from ODM and DoReMi computed with a 50k tokenizer Albalak et al. (2023). All baselines are trained under the same hardware and training budgets for fairness.

**Evaluation**   We report validation and test perplexity averaged over all domains. For downstream generalization, we evaluate on MMLU Hendrycks et al. (2021) with zero shot and five shot settings and on HumanEval Chen et al. (2021) with pass@1. These protocols are applied to models pretrained on The Pile and to models pretrained on SlimPajama under the same training recipe unless noted otherwise. In addition, for models pretrained on The Pile, we evaluate zero shot accuracy on five representative tasks that probe commonsense and scientific reasoning, namely COPA Roemmele et al. (2011), SciQ Welbl et al. (2017), LogiQA Liu et al. (2020), PIQA Bisk et al. (2020), and WinoGrande Sakaguchi et al. (2021). Together, these evaluations measure both language modeling quality and transfer to diverse downstream tasks.

## 4  FINDINGS AND DISCUSSION

### 4.1  MAIN RESULTS

Figure 1a shows results on The Pile: AC-ODM-410M, which trains a 1B target with a policy learned from a 410M proxy, reaches the ODM optimal validation perplexity with 71% fewer steps and, at 41,667 steps, achieves lower perplexity than TPW, ODM, and AC-ODM by 20.7%, 16.4%, and 13.1%. SlimPajama in Figure 3a exhibits the same pattern, where AC-ODM-410M requires 65% fewer steps than ODM and 73% fewer than the uniform baseline to reach its best perplexity and at

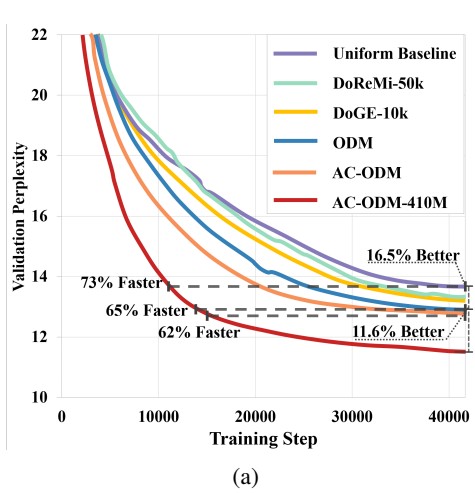
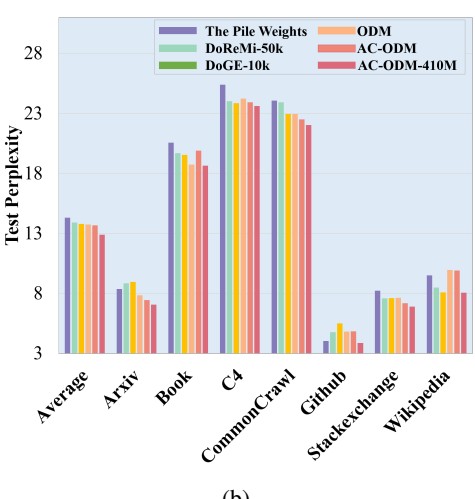

(a)                                                      (b)

Figure 3: **Results on SlimPajama with Pythia 1B. (a)** Validation perplexity during pretraining. AC-ODM and AC-ODM-410M converge faster than static and online baselines. AC-ODM-410M reaches the best perplexity of ODM in substantially fewer steps and yields lower perplexity at a fixed budget, consistent with the annotations. **(b)** Test perplexity averaged over domains and reported per domain. AC-ODM-410M attains the best average perplexity and is competitive or best across individual domains.

41,667 steps improves perplexity by $16.5\%$ over uniform and $11.6\%$ over the best online baseline. Overall, the proxy mode yields the strongest performance on both corpora, while the non proxy mode consistently improves over static and online baselines with negligible per step overhead.

AC-ODM's reward favors domains whose updates generalize to others, enabling the policy to exploit shared structure. On The Pile, Figure 4 shows that AC-ODM-410M attains the best average test perplexity and leads in most domains, improving by at least $20\%$ over ODM in 17 of 22 domains, while AC-ODM is best or near best elsewhere; gains are most pronounced in small and medium domains, and remain notable in the largest ones, indicating that the learned policy balances within domain learning with cross domain transfer. On SlimPajama, Figure 3b reports that AC-ODM-410M achieves the lowest average perplexity and is competitive or best across all seven domains, with smaller margins than on The Pile due to fewer domains and reduced opportunities for cross domain interactions. Together these results suggest that AC-ODM is particularly advantageous for large, finely partitioned corpora, while still improving convergence and final perplexity on smaller or coarser domain collections.

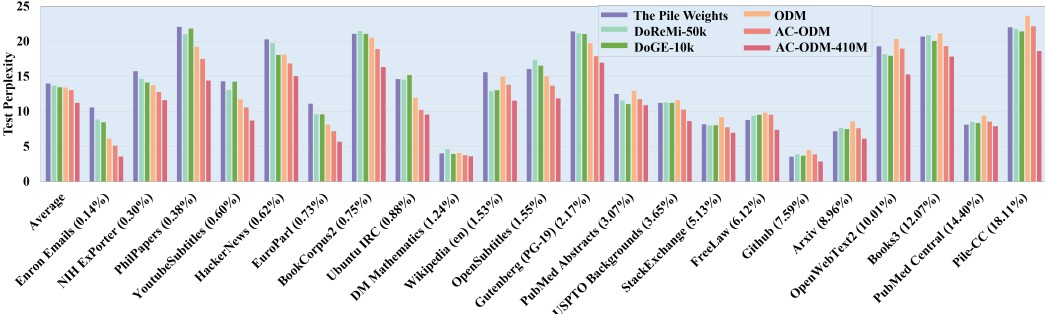

Figure 4: Test perplexity on average and on 22 individual domains of The Pile. The horizontal axis lists domain names with the corresponding proportion of tokens in the training set.

Table 1 summarizes results on MMLU and HumanEval. AC-ODM-410M improves over ODM by $27.5\%$ and $23.9\%$ on zero shot and five shot MMLU, and achieves a $2.23\times$ higher pass@1 on HumanEval. Figure 1b(b) complements these findings with additional zero shot evaluations on

Table 1: Evaluation of downstream tasks on MMLU and HumanEval. Acc denotes accuracy.

| Algorithm | MMLU 0 shot (Acc) | MMLU 5 shot (Acc) | HumanEval (pass@1) |
|---|---|---|---|
| TPW | 0.20664 | 0.27469 | 0.14119 |
| DoReMi-50k | 0.21862 | 0.27887 | 0.14215 |
| ODM | 0.23514 | 0.28416 | 0.32510 |
| AC-ODM | 0.25146 | 0.29868 | 0.60256 |
| AC-ODM-410M | **0.29980** | **0.35215** | **0.72644** |

COPA, SciQ, LogiQA, PIQA, and WinoGrande using the same The Pile pretrained checkpoints. The proxy mode consistently attains the highest accuracy across all five tasks, with large margins on COPA, SciQ, and PIQA, and steady gains on LogiQA and WinoGrande. The non proxy mode also improves reliably over ODM and TPW, which indicates that conditioning data mixing on the evolving training state is beneficial even without the proxy stage.

Across tasks in Figure 1b(b), AC-ODM tracks the proxy mode closely while adding negligible per step overhead, which makes it appealing when wall clock constraints dominate. AC-ODM-410M, which transfers the actor learned with a smaller proxy LLM, yields the strongest downstream performance, reflecting the value of learning the policy with explicit reward signals before guiding the target LLM. Complete per task comparisons with all baselines are provided in Appendix C.

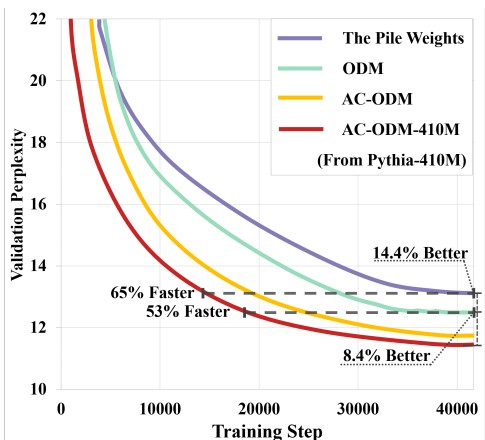

## 4.2 GENERALIZATION TO LLaMA STYLE ARCHITECTURES

To assess whether AC-ODM extends beyond Pythia, we repeat the pretraining study on a LLaMA style decoder only Transformer Dubey et al. (2024) with 0.9B parameters. As shown in Figure 5, AC-ODM improves the training dynamics of this modern architecture in the same way as for Pythia. The proxy mode remains the strongest:

Figure 5: Validation perplexity during LLaMA 0.9B pretraining on The Pile. We compare The Pile Weights, ODM, AC-ODM, and AC-ODM-410M, where the latter transfers an actor learned with a 410M Pythia proxy

it reaches a target validation perplexity with substantially fewer steps and achieves lower perplexity at a fixed budget. In particular, the annotations in Figure 5 indicate that AC-ODM-410M reduces the steps required to match a common perplexity level by about $65\%$ relative to The Pile Weights and by about $53\%$ relative to AC-ODM, and at the 41,667 step budget it improves perplexity by $14.4\%$ over The Pile Weights and by $8.4\%$ over AC-ODM.

The portability of the learned policy is noteworthy. The red curve in Figure 5 uses an actor trained once with a 410M Pythia proxy and then applied without modification to guide LLaMA pretraining. The resulting gains demonstrate that our state representation and reward are architecture agnostic and that the actor learned on one family can effectively guide another family with similar decoder only structure. This cross model transfer, together with the strong performance of the non proxy mode, supports the conclusion that AC-ODM is effective for modern LLMs and can generalize across closely related architectures.

## 4.3 COMPUTATIONAL COST

We compare the compute required by AC-ODM and ODM to train a 1B LLM to the validation perplexity achieved by ODM under identical hardware. Direct AC-ODM adds only $0.4\%$ per step overhead (2.48 vs. 2.47 s) but reduces steps by $31.95\%$ ($41667 \rightarrow 28356$), giving a $1.46\times$ end to end speedup. In the proxy mode, the actor–critic is learned on a smaller proxy and then transferred to the 1B target, which then needs only $28.82\%$ of the ODM steps ($41667 \rightarrow 12500$ or $12010$).

Table 2: Model size and computational cost during pretraining. The columns AC, LLM, and AC+LLM denote the parameter counts of the actor–critic networks, the language model, and the combined system. For proxy configurations, the rows labeled AC-ODM(160M) and AC-ODM(410M) report the steps used to converge the actor–critic on the proxy, while the rows labeled AC-ODM(1B) report the steps taken by the 1B target to match the ODM perplexity.

| Algorithm | AC | LLM | AC+LLM | Time per step (s) | Steps | Speedup ratio |
|---|---|---|---|---|---|---|
| ODM | 0 | 1B | 1B | **2.47** | 41667 | 1x |
| AC-ODM | 17.32M | 1B | 1.02B | **2.48** | 28356 | 1.46x |
| AC-ODM(160M) | 17.32M | 160M | 177.32M | 0.65 | 28690 | **2.08x** |
| AC-ODM(1B) | 17.32M | 1B | 1.02B | 2.48 | 12500 | |
| AC-ODM(410M) | 17.32M | 410M | 427.32M | 1.41 | 28690 | **1.47x** |
| AC-ODM(1B) | 17.32M | 1B | 1.02B | 2.48 | 12010 | |

Counting the proxy stage, the overall speedup is $2.08\times$ with a 160M proxy and $1.47\times$ with a 410M proxy. Although the larger proxy increases pretraining cost, the stronger policy it learns amortizes over larger targets, making the proxy mode increasingly attractive at scale.

### 4.4 SUBEFFECT OF PROXY MODEL SIZE

Figure 6 compares a 1B target trained with sampling policies learned from 70M, 160M, and 410M proxy LLMs against joint training with AC-ODM. The proxies attain training losses of 2.8, 2.65, and 2.48, with validation perplexities 20.3, 15.5, and 12.1, respectively. Policies from 160M and 410M consistently outperform joint AC-ODM, indicating that a prelearned actor adapts from the first step, whereas an online actor is still converging. The 70M proxy performs worst, suggesting insufficient capacity to learn a transferable policy. The 160M proxy nearly matches the 410M proxy, especially early in training, likely because the 1B target limits headroom. We expect the gap to widen for larger targets and leave a systematic study of proxy–target scaling for future work.

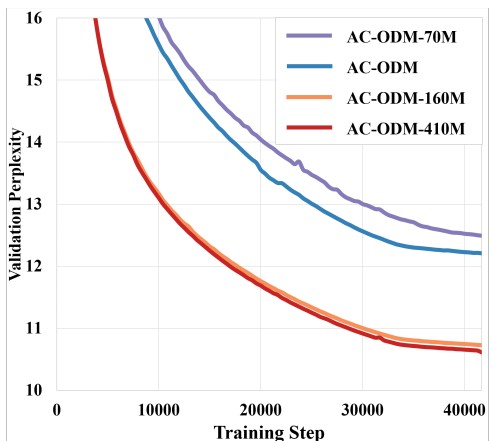

Figure 6: Validation perplexity for a 1B target using policies learned with proxy LLMs of different sizes (average over 22 Pile domains).

## 5 CONCLUSION

We presented AC-ODM, an actor and critic online data mixing method that treats LLM pretraining as reinforcement learning and optimizes a gradient alignment reward to capture intra domain interactions. Both the non proxy and proxy modes are effective, with the proxy mode strongest. On The Pile and SlimPajama, a 410M proxy policy enables a 1B target to reach the ODM optimum in 71% and 65% fewer steps, and at 41,667 steps lowers perplexity relative to TPW, ODM, and AC-ODM by 20.7%, 16.4%, and 13.1%, with only 0.4% per step overhead. Downstream, AC-ODM-410M improves zero shot and five shot MMLU by 27.5% and 23.9%, achieves a $2.23\times$ higher pass@1 on HumanEval, and yields an average $+3.44\%$ accuracy gain across COPA, SciQ, LogiQA, PIQA, and WinoGrande. On a LLaMA style 0.9B model, AC-ODM maintains the fastest convergence, and an actor learned on a 410M Pythia proxy transfers effectively to LLaMA, indicating cross model portability. These results suggest AC-ODM is especially beneficial for large and finely partitioned corpora.

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

Table 3: Ablation study of selected layers used for reward computation.

| Block indexes | Perplexity |
|---|---|
| 12, 14, 16 | 13.0655 |
| 14, 15, 16 | 13.0682 |
| 6, 8, 10 | 13.0709 |
| 1, 2, 3 | 13.0701 |

Table 4: Zero shot accuracy on downstream tasks using The Pile pretrained 1B models. AVG is the macro average across tasks.

| Task | TPW | DoReMi-50k | DoGE-10k | ODM | AC-ODM | AC-ODM-410M |
|---|---|---|---|---|---|---|
| MMLU | 0.27469 | 0.27887 | 0.27955 | 0.28416 | 0.29868 | **0.35215** |
| COPA | 0.54800 | 0.62000 | 0.64800 | 0.68000 | 0.69800 | **0.72000** |
| SciQ | 0.62000 | 0.63800 | 0.66400 | 0.68900 | 0.70200 | **0.73000** |
| LogiQA | 0.23810 | 0.24580 | 0.27650 | 0.30720 | 0.30110 | **0.32260** |
| PIQA | 0.60330 | 0.61670 | 0.62000 | 0.68330 | 0.69670 | **0.72000** |
| WinoGrande | 0.50930 | 0.52630 | 0.53200 | 0.59650 | 0.58300 | **0.63380** |
| AVG | 0.50374 | 0.52936 | 0.54810 | 0.59120 | 0.59616 | **0.62528** |

# A    APPENDIX

## A.1    LLM MODEL CONFIGURATION

We adopt the sequence length of 1024 and employ a 16-layer Transformers architecture with a hidden size of 2048 and 16 attention heads. Rotary positional embedding Su et al. (2023) is incorporated. We leverage FlashAttention Dao et al. (2022), which optimizes memory access and reduces computation overhead, to improve training efficiency. The model is trained using Adam optimizer Kingma & Ba (2017). The learning rate undergoes a linear warm-up for 833 iterations, gradually increasing from a minimum of 2.5e-5 to a peak of 2.5e-4, followed by a cosine decay back to 2.5e-5. We utilize the GPT-NeoX-20B tokenizer Black et al. (2022) for text processing.

## A.2    AC NETWORKS CONFIGURATION

For both the actor and the critic, we employ a fully connected 6-layer neural network with 1024 neurons per hidden layer. Except for the output layer, each layer is followed by layer normalization and a ReLU activation. In the actor, the output layer is further processed by the softmax activation function, while in the critic, the output layer is post-processed by the identity activation function.

# B    APPENDIX

The results show that using later Transformer blocks yields the best proxy for reward computation: selecting layers 12,14,16 attains the lowest perplexity, slightly outperforming contiguous later layers 14,15,16 and clearly matching or exceeding mid and early layer choices. Although the absolute differences are small, they are consistent, suggesting that mid-to-late representations provide a more informative signal while confirming that AC-ODM is robust to the exact layer subset. These findings support our default choice of 12,14,16.

# C    APPENDIX

**Analysis.** AC-ODM-410M achieves the highest accuracy on every task and the best average (0.62528), improving over ODM by an absolute $+0.0341$ and a relative $+5.8\%$. Gains are consistent across commonsense and reasoning benchmarks, with the largest jump on MMLU. The non

Table 5: Ablation study of state components used by AC-ODM. Removing any component degrades performance; "Impr." is the relative change in perplexity compared with using all components.

| Status of state | Perplexity | Impr. |
|---|---|---|
| All components | 13.0655 | – |
| w/o $n$: number of samples per domain | 13.1203 | $-0.419\%$ |
| w/o $t$: iteration step | 13.0958 | $-0.232\%$ |
| w/o $\ell(\theta_M, B)$: per-domain losses | 13.8992 | $-6.38\%$ |
| w/o $\Delta\ell(\theta_M, B)$: change of per-domain losses | 13.5470 | $-3.69\%$ |
| w/o $\|\omega\|_2$: $L^2$ norm of selected layer weights | 13.9115 | $-6.48\%$ |
| w/o $\|\Delta\omega\|_2$: $L^2$ norm change of selected layers | 13.4254 | $-2.75\%$ |

proxy AC-ODM also improves over ODM on average but trails the proxy mode, underscoring the benefit of learning the policy with a proxy model before guiding the target LLM.

## D  APPENDIX

**Analysis.** All six features contribute to policy quality. Removing per-domain losses $\ell(\theta_M, B)$ or the weight norm $\|\omega\|_2$ causes the largest degradations ($\approx 6.4\%$), indicating that absolute training signal and model-scale dynamics are critical for the actor. The change-of-loss term $\Delta\ell(\theta_M, B)$ is also important ($-3.69\%$), while the count of seen samples $n$ and the step index $t$ provide smaller but nontrivial gains. Overall, the full state offers the best perplexity and each component carries complementary information.

## E  EVOLUTION OF DOMAIN WEIGHTS DURING TRAINING

Figure 7a–7d illustrate the evolution of domain weights across 22 distinct domains in The Pile dataset during training 1B Pythia model under AC-ODM algorithm. AC-ODM initializes from the original domain weights of The Pile and undergoes dynamic updates during the warmup phase. After approximately 15,000 training steps, the domain weights stabilize. Afterward, minor fluctuations are observed, which correspond to the evolving state of the LLM. The adaptive nature of AC-ODM's domain weight generation during this critical phase allows it to better align with the evolving model state, thereby facilitating faster reductions in both training loss and perplexity compared to prior methods.

Both AC-ODM and ODM algorithms eventually converge to stable domain weights. However, AC-ODM exhibits more substantial adjustments in domain weights during the first third of training, while ODM Albalak et al. (2023) stabilizes after only the first fifth of the total training steps. Notably, even after reaching stability, AC-ODM continues to experience slight fluctuations in domain weights, enabling dynamic adaptation to evolving LLM state. In contrast, domain weights in ODM remain nearly constant in the later stages of training, indicating a lack of flexibility in response to parameter updates in later stage.

A comparison of domains with the large magnitudes of increases or decreases in weights across Figure 7a–7d reveals consistent patterns. Regardless of the token proportion, domains characterized by high-quality and general-purpose texts tend to experience weight increases during training. Examples include HackerNews in Figure 7a, Gutenberg (PG-19) and BookCorpus2 in Figure 7b, Stack-Exchange and USPTO Backgrounds in Figure 7c, and Book3 in Figure 7d. In contrast, domains containing noisier texts or highly domain-specific contents exhibit significant weight reductions, such as Enron Emails in Figure 7a, DM Mathematics and Wikipedia (en) in Figure 7b, Github and FreeLaw in Figure 7c, and PubMed Central in Figure 7d. These observations align with human intuitive expectations: during LLM pretraining, data domains rich in high-quality, generalizable content are more effective at driving model convergence in the early stages of training.

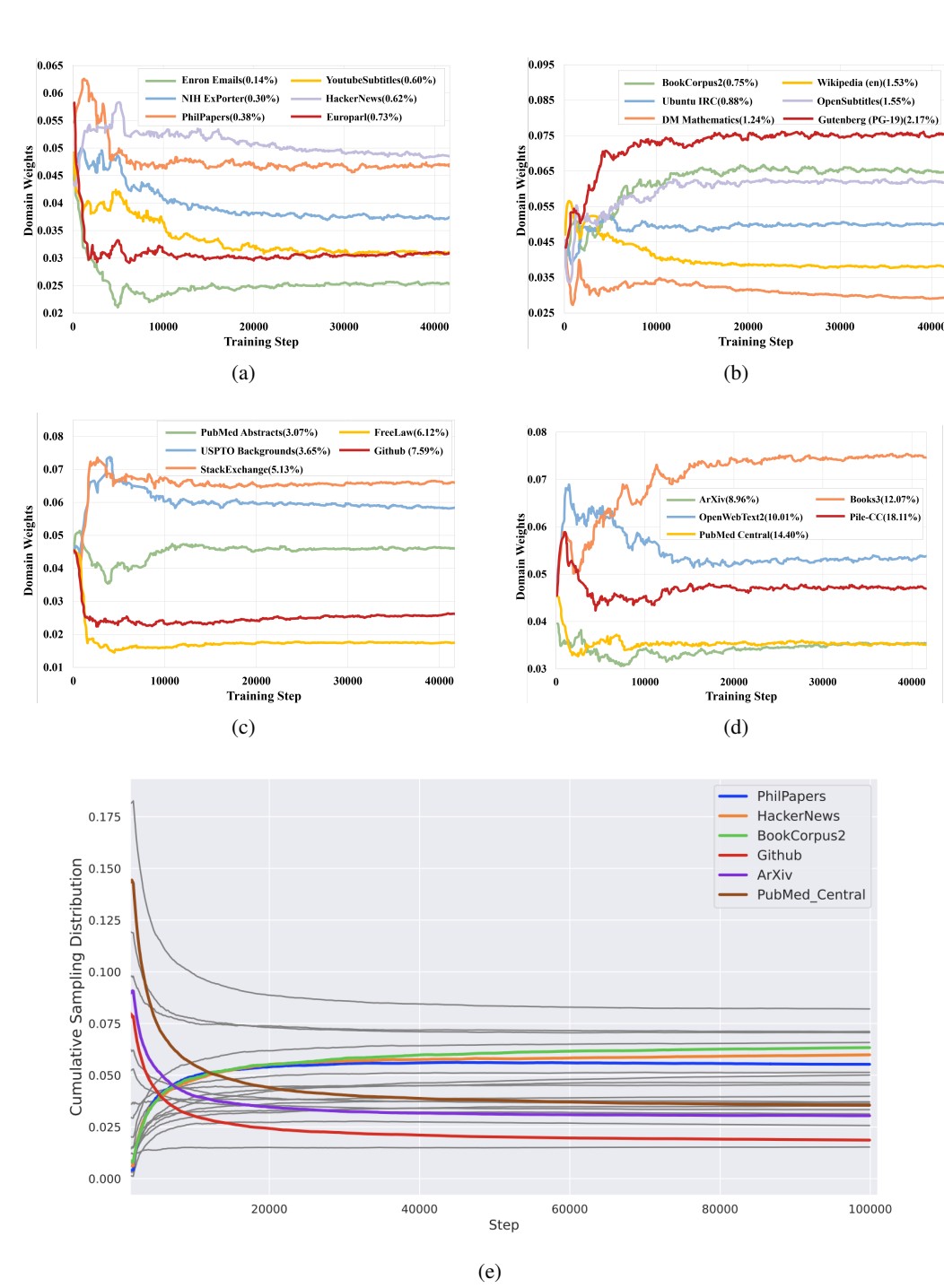

Figure 7: Evolution of domain weights during training. The legend indicates the proportion of tokens of each domain (in percentage). (a) Six domains with smallest token proportions; (b) Six domains with token proportions below 3%; (c) Five domains with token proportions below 8%; (d) Five domains with highest token proportions; (e) The cumulative sampling distribution of ODM Albalak et al. (2023).

Table 6: Zero-shot accuracy of AC-ODMs among different groups in MMLU.

| Algorithm | STEM | Social Sciences | Humanities | Other | Average |
|---|---|---|---|---|---|
| AC-ODM | 0.24213 | 0.30433 | 0.25381 | 0.24626 | 0.25146 |
| AC-ODM-410M | 0.28219 | 0.38231 | 0.29908 | 0.28924 | 0.29980 |

# F    ANALYSIS OF RESULTS OF MMLU TASKS

We evaluate the performance of AC-ODM across four domain-specific groups in the MMLU benchmark, along with the overall average accuracy. As shown in Table 6, AC-ODM achieves better accuracy in the *Social Sciences* group, achieving approximately 21% higher than average. This indicates that AC-ODM effectively adapts to domain shifts in this group, likely benefiting from the alignment between Social Sciences content and the training distribution in The Pile. In contrast, AC-ODM underperforms in the *STEM* and *Other* groups, where accuracy falls slightly below the overall average. The *Humanities* group yields performance close to the average. These observations suggest that AC-ODM facilitates the LLM's ability to better acquire and generalize semantic patterns related to humanities and social science domains from The Pile.

Compared to the direct application of AC-ODM, the proxy-based AC-ODM-410M variant consistently improves performance across all groups, yielding an overall 19% increase in average accuracy. The most notable gains occur in the *Social Sciences* and *Other* groups, with improvements of 26% and 17%, respectively. These results indicate that AC-ODM trained on a 410M-parameter proxy model can effectively capture the underlying domain relationships present in The Pile, which are transferable to larger models and particularly beneficial for tasks involving humanities, social sciences, and general knowledge. However, the relatively limited gains in STEM-related domains also suggest that AC-ODM pays less attention to exploring domain-specific features relevant to science and engineering. This limitation may stem from the relatively low proportion of STEM-related content in The Pile dataset itself, which we would like to investigate in the future.

Figure 8 illustrates the task-level accuracy of AC-ODMs across different groups within the MMLU benchmark. In the *STEM* group, AC-ODMs achieve strong performance on tasks such as *Electrical Engineering* and *Computer Security*. Within the *Social Sciences* group, notable improvements are observed in *US Foreign Policy*, *Professional Psychology*, *High School Psychology*, and *Econometrics*. For the *Humanities* group, AC-ODMs perform well on *World Religions*, *Logical Fallacies*, and *Jurisprudence*. In the *Other* group, tasks such as *Marketing*, *Human Aging*, *College Medicine*, and *Clinical Knowledge* benefit significantly from AC-ODMs. These results suggest that AC-ODM's domain weight optimization strategy effectively guides the LLMs to acquire semantic information associated with general-purpose knowledge domains.

Compared to AC-ODM, the proxy-based AC-ODM-410M consistently improves performance across all tasks. Notably, for particularly challenging tasks such as *High School Statistics*, *Elementary Mathematics*, and *Management*, AC-ODM-410M achieves non-zero accuracy where AC-ODM fails completely (0% accuracy). These findings highlight that the use of a well-trained proxy model during training enables AC-ODM to capture meaningful domain relationships, ultimately enhancing LLM performance. Proxy-based training allows the model to better infer the latent structure of domain-specific knowledge while fulfilling difficult tasks, thereby leading to more effective adaptation and improved generalization.

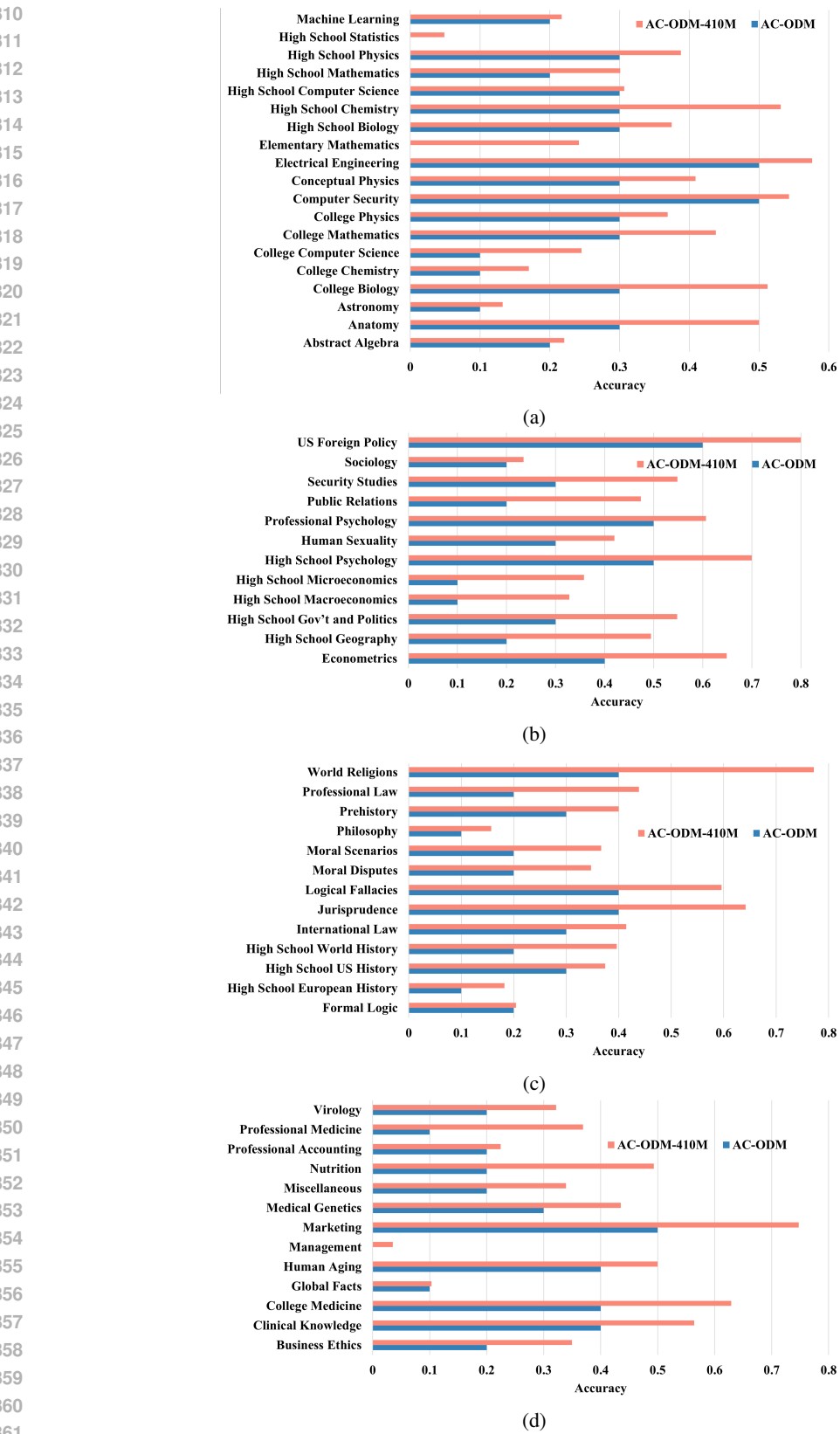

Figure 8: Zero-shot accuracy of AC-ODMs across MMLU tasks, grouped by subject category. (a) STEM; (b) Social Sciences; (c) Humanities; (d) Other.