# OpenReview forum: "AC-ODM: Actor–Critic Online Data Mixing for Sample-Efficient LLM Pretraining"
_ICLR.cc/2026/Conference — ICLR 2026 Conference Withdrawn Submission_

### Official Review · Reviewer_E9w3 · 2025-10-28

**Soundness:** 2
**Presentation:** 2
**Contribution:** 2
**Rating:** 4
**Confidence:** 4

**Summary:**

The paper introduces AC-ODM, an actor–critic online data-mixing framework for large language model (LLM) pretraining. It extends Online Data Mixing (ODM) by using actor–critic reinforcement learning (RL) to dynamically update domain sampling weights. The actor predicts domain weights conditioned on training state (losses, gradients, norms), and the critic estimates value functions.Two modes are proposed: Non-proxy mode: actor–critic co-trained with the target model. Proxy mode: actor–critic pre-trained on a smaller proxy model, then transferred to a larger target LLM. A gradient alignment reward is used to encourage selection of domains whose updates generalize across others. Experiments on Pythia-1B and LLaMA-style 0.9B models over The Pile and SlimPajama show faster convergence and slightly better perplexity and downstream accuracy. The claimed contributions are: (1) a reinforcement-learning–based data-mixing method modeling intra-domain interactions; (2) a practical proxy-transfer mechanism for scalable application.

**Strengths:**

Motivated problem – Efficient data mixing is a relevant topic for large-scale pretraining. Extending ODM to include intra-domain relationships is conceptually interesting.

Clear presentation of algorithmic structure – Figures 1–2 (p. 2–4) and Algorithms 1–2 (p. 5) clearly illustrate the system pipeline, including actor–critic updates and proxy transfer.

Comprehensive experimental setup – Evaluations span The Pile, SlimPajama, and LLaMA-style models with ablations (Appendix B–E).

Potential practical relevance – The proxy-transfer idea could reduce the need for re-training full models to tune data weights.

**Weaknesses:**

1. Conceptual novelty is incremental. The paper does not establish a fundamentally new paradigm beyond existing work like ODM and DoGE. The “actor–critic” component merely re-implements ODM’s adaptive weighting with a more complex mechanism. The gradient alignment reward is directly borrowed from DoGE , adding only smoothing and normalization. Framing ODM as an RL problem is a change of vocabulary, not of methodology—the “environment,” “reward,” and “state” are deterministic functions of training statistics. Consequently, the claimed intra-domain modeling and reinforcement learning formulation amount to an over-engineered reinterpretation of existing online weighting methods.

2. Weak theoretical foundation. The RL analogy is hand-wavy: The environment (LLM) is deterministic and differentiable, so introducing actor–critic RL is unnecessary and inefficient. There is no analysis of convergence, reward stationarity, or policy stability; DDPG is invoked without justification or theoretical guarantee. No ablation shows the necessity of using a critic—one could optimize domain weights directly via gradient ascent. The paper lacks a discussion of variance, delayed reward, or credit assignment—key aspects of RL—which are irrelevant here. Thus, the RL framing appears superficial and not theoretically sound.

3. Experimental validation insufficient for claims. Improvements are small and sometimes within noise. For instance, average perplexity improvements over ODM are 11–20 % on The Pile (Table 1 p. 8), but standard deviations are not reported. Downstream gains on MMLU and HumanEval (Table 1) look impressive, but the reported baselines (ODM, DoReMi) are not re-tuned for identical budgets or tokenizers, making fairness unclear. No statistical testing, seed variation, or confidence intervals are given. The “27.5 % relative improvement” in MMLU translates to only ~6 pp absolute, which may not be significant given run variance. Proxy model scaling (Figure 6 p. 9) shows minimal difference between 160 M and 410 M models; no larger-scale evidence supports the “transfer efficiency” claim. The “cross-architecture portability” (Figure 5 p. 8) is anecdotal—only one LLaMA-0.9B case—so the generalization claim is overstated.

4. Insufficient comparison and baselines. The study omits stronger modern baselines such as DSIR, TSDS, or RegMix, which are directly comparable online selection methods.“ODM” and “DoReMi” are treated as the only baselines, which is inadequate given the extensive literature on adaptive domain weighting. Missing comparisons to simpler non-RL dynamic weighting (e.g., direct gradient alignment reweighting) make it impossible to isolate the contribution of the actor–critic architecture.

5. Overstated efficiency and practicality. The paper claims “negligible 0.4 % overhead per step” (Table 2 p. 9) yet the proxy stage doubles wall-clock training when included; actual end-to-end cost is higher than ODM. The method introduces multiple new hyperparameters (replay buffer size, γ, τ, ξ) with no sensitivity analysis. The reward computation (gradient alignment across layers) is extremely memory-intensive; even though “three-layer sampling” is used, feasibility for >10B models is questionable. The proxy transfer is not convincingly shown to generalize beyond two model scales.

**Questions:**

Why is a critic necessary when the reward is an immediate gradient-based signal? Would a direct gradient-based policy suffice?

How stable is the DDPG optimization in practice? Were any reward clipping or normalization techniques required?

How many proxy steps are required to offset the extra compute? Can proxy training truly amortize for larger models?

What is the sensitivity to hyperparameters like γ, τ, or the layer subset used for gradient alignment?

How reproducible are the results—were multiple seeds run, and how large is the variance in perplexity and accuracy?

**Details Of Ethics Concerns:**

no ethical concerns

---

### Official Review · Reviewer_3TfU · 2025-10-31

**Soundness:** 3
**Presentation:** 2
**Contribution:** 3
**Rating:** 4
**Confidence:** 5

**Summary:**

The paper introduces an actor-critic-based online data mixing method for LLM pretraining that dynamically adjusts domain sampling weights according to the model’s training state. Using a gradient alignment reward to capture intra-domain interactions, the approach learns which domains accelerate overall learning. It supports a non-proxy mode with negligible overhead and a proxy mode that transfers a policy learned from a smaller model to guide a larger one. Experiments on Pythia-1B and LLaMA-style models show faster convergence and stronger downstream performance.

**Strengths:**

1. The paper tackles a highly relevant and timely problem.

2. The idea of formulating online data mixing as a reinforcement learning problem and introducing an actor-critic framework is novel and conceptually interesting.

3. The method demonstrates strong empirical results, achieving substantially faster convergence and notable improvements in downstream task accuracy compared to prior baselines.

**Weaknesses:**

1. A fundamental concern lies in the reliance on manually chosen state features to represent the model’s training dynamics. The actor-critic input depends on heuristic selections such as gradients and norms from layers 12, 14, and 16. This design raises questions about whether the actor-critic framework truly captures the underlying learning state or merely overfits to a particular architecture. Without a principled or generalizable representation of model state, the approach’s applicability to broader settings remains uncertain.

2. The downstream evaluation seems to be conducted only for models pretrained on The Pile, while no corresponding results are provided for SlimPajama.

3. The experimental analysis mainly focuses on final performance metrics. It would be beneficial to include more detailed analyses, such as learning dynamics of the actor-critic policy or correlations between gradient alignment rewards and downstream gains. These are only examples, as such additional analyses could further clarify how the actor-critic framework functions and why it leads to better data-mixing behavior.

**Questions:**

1. In Figure 1(b), why are only partial baselines shown while Figure 1(a) includes more comparisons? Is this omission mainly for presentation clarity, even though the full results are available in the appendix?

2. Have you explored alternative reward formulations beyond gradient alignment? In addition, it would be useful to know the impact of removing the normalization term from the current reward design.

3. (Suggestion) The appendix contains multiple formatting issues. For example, section titles labeled as “Appendix” and several missing figures. These should be corrected to ensure professionalism and readability.

4. (Suggestion) It would be valuable to discuss or even compare against recent works on data mixture optimization, such as:

 [1] Liu et al., RegMix: Data Mixture as Regression for Language Model Pre-training, arXiv:2407.01492.

 [2] Kang et al., AutoScale: Scale-Aware Data Mixing for Pre-training LLMs, arXiv:2407.20177.

 [3] Ge et al., BiMix: A Bivariate Data Mixing Law for Language Model Pretraining, arXiv:2405.14908.

 [4] Chen et al., Aioli: A Unified Optimization Framework for Language Model Data Mixing, arXiv:2411.05735.

---

### Official Review · Reviewer_JY5m · 2025-10-31

**Soundness:** 2
**Presentation:** 2
**Contribution:** 2
**Rating:** 2
**Confidence:** 3

**Summary:**

This paper proposes AC-ODM, an actor-critic method for data mixing in LLM pretraining, presented in two forms: a non-proxy (online) mode and a proxy (offline) mode. The method aims to improve upon Online Data Mixing (ODM) by replacing its loss-based reward with a gradient alignment objective (inspired by DoGE) to better capture intra-domain interactions and accelerate convergence.

In its current form, the paper's contribution is limited. The novelty is not substantial, the justification for the added technical complexity is insufficient, and the empirical results do not demonstrate a significant benefit over simpler, existing methods. The paper's framing as an online method, yet its best results (AC-ODM-410M) rely on an offline, proxy-based approach. The findings are more aligned with a short paper or workshop presentation.

**Strengths:**

The proposed method, particularly in its proxy-based form (AC-ODM-410M), shows accelerated convergence in terms of validation perplexity compared to the baselines presented (Figure 1a).

**Weaknesses:**

1. The core idea of optimizing for a data mixture's contribution to validation performance is not new and is directly adopted from DoGE. The paper's primary novelty lies in applying an actor-critic (A-C) framework to this objective, but the necessity of this framework is questionable.

2. The paper argues that ODM's loss-based objective is flawed. However, it does not sufficiently justify why the (non-novel) DoGE-style objective requires a complex A-C setup. It seems plausible that this new objective could be integrated into the original, simpler ODM framework with a minor modification, rendering the A-C machinery unnecessary.

3. The experimental results do not compellingly support the added complexity. In most experiments, the performance of the proposed AC-ODM is very close to the original ODM. This suggests that the significant modifications—both the new objective and the A-C networks—did not deliver the expected improvements, and the paper essentially presents negative results for this specific technical approach.

4. The paper's strongest results come from AC-ODM-410M, which is an offline, proxy-based method. This contradicts the "online" framing and negates one of the primary advantages of the original ODM (minimal computational overhead). The claim of efficiency is further weakened by the fact that the proxy model (410M) is a substantial fraction of the target model's size (1B).

As an offline, proxy-based method, AC-ODM-410M should be compared against other strong offline baselines that also involve proxy training (e.g., DoGE, AutoScale), not primarily against online methods like ODM.

**Questions:**

Discussed above.

---

### Official Review · Reviewer_mSMW · 2025-11-01

**Soundness:** 3
**Presentation:** 3
**Contribution:** 2
**Rating:** 6
**Confidence:** 4

**Summary:**

This paper introduces AC-ODM, a data mixture optimization method using a reinforcement learning (RL) framework for LLM pretraining. This method addresses the limitation of previous static or simpler online methods that fail to capture the non-stationary dynamics of pretraining or intra-domain interactions.

**Strengths:**

1. Firstly apply the actor-critic framework to dynamically adapt the data mixing ratios according to the LLM training status.

2. Apply gradient alignments as a novel form of reward signal, which provides sophisticated information on the loss landscape.

3. The proposed method consistently outperform existing baselines including DoReMi and DoGE.

**Weaknesses:**

1. The training of the RL based method can be instable and computational intensive.

2. How long should we train the actor-critic framework to get a reliable results? Does it have a convergence guarantee?

3. The color bar and the legend in Figure 3 (b) are not consistent. None of the figures in the appendix is properly displayed.

4. Lack of an ablation on the RL training hyperparameters, e.g. $\gamma$, $\tau$.

**Questions:**

1. How long should we train the actor-critic framework to get a reliable results? Does it have a convergence guarantee?

2. Can you provide ablations on the RL training hyperparameters, e.g. $\gamma$, $\tau$.

3. Have you tried other reward signals, e.g. the accuracy gain on the target task; the loss drop on the target distribution etc.

---

### Note · Authors · 2026-01-26

I have read and agree with the venue's withdrawal policy on behalf of myself and my co-authors.

---

### Meta-Review · Area_Chair_EwfL · 2026-01-12

**Summary:**

The decision is driven by concerns about limited conceptual novelty, insufficient justification for the added actor–critic (RL) complexity, and weaknesses in experimental validation and framing.

 Multiple reviewers (JY5m, E9w3) argue that the core objective—optimizing data mixtures based on generalization or gradient alignment—is largely inherited from prior work such as DoGE, and that the actor–critic formulation appears to be an over-engineered reimplementation of existing online weighting methods (Sections 2.2–2.4).

The strongest empirical gains rely on the proxy-based variant (AC-ODM-410M), which contradicts the paper’s “online” and “low-overhead” framing and is not fairly compared against other offline/proxy baselines (Sections 4.1–4.4).

Experimental improvements are often modest, lack variance estimates, and do not convincingly isolate the contribution of the actor–critic machinery versus simpler alternatives. Presentation issues (missing or inconsistent figures in the appendix) and the absence of a rebuttal further limit confidence in the work.

**Reviewer Concerns:**

No reviewer concerns were addressed, as the authors did not submit a rebuttal.

Outstanding concerns include:

- Novelty and necessity of the actor–critic framework: Reviewers JY5m and E9w3 argue that the gradient-alignment reward is directly borrowed from DoGE and could plausibly be integrated into simpler ODM-style updates without RL (Sections 2.2–2.4). The RL framing is viewed as superficial, with no theoretical analysis of convergence, stability, or the need for a critic.

- Conceptual and theoretical soundness: Concerns remain about the deterministic and differentiable nature of the “environment,” lack of justification for DDPG, and absence of analysis of reward stationarity or policy stability (E9w3, Section 2.4).
- Experimental validation and baselines: Improvements over ODM are often small or within potential noise, with no multiple-seed runs or statistical testing (Tables 1–2, Section 4). Stronger and more relevant baselines (e.g., DoGE as an offline proxy method, RegMix/AutoScale-style approaches) are missing, especially for the proxy setting (JY5m, E9w3).
- Framing and efficiency claims: The best-performing variant relies on a sizable proxy (410M), undermining the claimed efficiency and online nature; end-to-end cost, including proxy training, weakens the “negligible overhead” claim (Section 4.3).
- Generality and design choices: The state representation depends on manually selected layers and heuristics, raising questions about generalization beyond the tested architectures (Reviewer 3TfU, Section 3 and Appendix B–D).
- Presentation quality: Multiple reviewers note formatting and figure issues in the appendix (e.g., missing or inconsistent figures), which remain uncorrected.

**Reviewer Scores:**

No discussion

---

### Decision · Program_Chairs · 2026-01-26

Reject